# The Influence of Nicotine on Trophoblast-Derived Exosomes in a Mouse Model of Pathogenic Preeclampsia

**DOI:** 10.3390/ijms241311126

**Published:** 2023-07-05

**Authors:** Ayane Kubo, Keiichi Matsubara, Yuko Matsubara, Hirotomo Nakaoka, Takashi Sugiyama

**Affiliations:** 1Department of Obstetrics and Gynecology, Ehime University Graduate School of Medicine, Shitsukawa, Toon 791-0295, Ehime, Japan; h401025x@mails.cc.ehime-u.ac.jp (A.K.); takeyu@m.ehime-u.ac.jp (Y.M.); sugiyama@m.ehime-u.ac.jp (T.S.); 2Department of Regional Pediatrics and Perinatology, Ehime University Graduate School of Medicine, Shitsukawa, Toon 791-0295, Ehime, Japan; 3Advanced Research Support Center, Ehime University Graduate School of Medicine, Shitsukawa, Toon 791-0295, Ehime, Japan; hnakaoka@m.ehime-u.ac.jp

**Keywords:** preeclampsia, exosomes, nicotine, proteomics, bioinformatics

## Abstract

Preeclampsia (PE) is a serious complication of pregnancy with a pathogenesis that is not fully understood, though it involves the impaired invasion of extravillous trophoblasts (EVTs) into the decidual layer during implantation. Because the risk of PE is actually decreased by cigarette smoking, we considered the possibility that nicotine, a critical component of tobacco smoke, might protect against PE by modifying the content of exosomes from EVTs. We investigated the effects of nicotine on our PE model mouse and evaluated blood pressure. Next, exosomes were extracted from nicotine-treated extravillous trophoblasts (HTR-8/SVneo), and the peptide samples were evaluated by DIA (Data Independent Acquisition) proteomic analysis following nano LC-MS/MS. Hub proteins were identified using bioinformatic analysis. We found that nicotine significantly reduced blood pressure in a PE mouse model. Furthermore, we identified many proteins whose abundance in exosomes was modified by nicotine treatment of EVTs, and we used bioinformatic annotation and network analysis to select five key hub proteins with potential roles in the pathogenesis or prevention of PE. EVT-derived exosomes might influence the pathogenesis of PE because the cargo delivered by exosomes can signal to and modify the receiving cells and their environment.

## 1. Introduction

Preeclampsia (PE) is a serious, life-threatening disease that afflicts hypertensive pregnant women with symptoms that can include proteinuria, headaches, fetal growth restriction (FGR), liver and kidney dysfunction, seizures, and abnormal blood tests for coagulation and fetal factors. The development of PE often improves following delivery and may involve placental-derived factors, and a two-stage theory has been proposed to explain the pathogenesis [1,2] (Figure 1). In the first stage, extravillous trophoblasts (EVTs), which invade the uterine endometrium and myometrium in the first trimester of normal pregnancy, become functionally impaired, leading to disturbed remodeling of the spiral artery and poor placentation with uteroplacental ischemic defects. In the second stage, this dysfunctional placenta releases humoral factors into the maternal systemic circulation that lead to multi-organ damage and the hallmark symptoms of the disease. These factors, which include proinflammatory cytokines (e.g., tumor necrosis factor-alpha (TNFα), and interferon-gamma (IFN-γ)) and vasoactive substances (e.g., endothelin-1 (ET-1), thromboxane A2, and angiotensin II), can increase vasoconstriction, vascular permeability, and coagulopathy [3,4,5,6].

EVTs can proliferate and infiltrate in the decidua and are directly involved in the remodeling of spiral arteries; however, Salomon et al. [7] proposed an alternative pathway in which the humoral factors contained within EVT exosomes might indirectly promote spiral artery remodeling by stimulating the proliferation and invasion of vascular smooth muscle cells. Exosomes are extracellular vesicles of approximately 100 nm in diameter that are secreted by various cell types; they are bounded by a lipid bilayer and contain RNA, DNA, and proteins cargo. Changes in exosome contents might promote the pathogenesis of PE as a result of impaired remodeling of spiral arteries; indeed, miRNA profiles in the plasma exosomes of PE patients have been reported [8]. Furthermore, the mechanism by which EVTs influence the uterine tissue and distant organs is thought to involve exosomes, which are secreted extracellular vesicles loaded with molecular cargo (e.g., DNA, RNA, proteins), that travel to and communicate with recipient cells. These exosomes carry signals to create a local environment suitable for EVT invasion, proliferation, and remodeling of the spiral artery, and these vesicles are also distributed more broadly to distant maternal organs in a strategy akin to that of cancer cell proliferation and invasion [9]. We are interested in the possibility that PE is initiated and maintained by the abnormal secretion and composition of EVT exosomes and therefore sought to examine exosome cargo for clues to the mechanism of influence on PE. To develop an experimental system in which the relationship between exosomes and PE pathogenesis can be manipulated, we took advantage of the observation that PE risk is reduced by tobacco smoking even though it increases other pregnancy risks (e.g., fetal growth restriction and placental abruption) [10,11]. Because nicotine is a critical bioactive component of tobacco smoke that accumulates quickly in and affects many organs [12,13,14], we focused here on the possibility that nicotine acts on EVTs to modify exosome secretion and cargo composition in the context of PE pathogenesis. Tyagi et al. reported that chronic exposure of lung cancer patients to nicotine stimulates neutrophils to secrete miR-4466-rich exosomes, which are involved in the process of tumor metastasis [15]. Nicotine also stimulates macrophages to secrete miR-21-3p-rich exosomes, which act on vascular smooth muscle and promote atherosclerosis [16]. Taken together, these studies indicate that nicotine contributed to various pathological conditions by stimulating the secretion of exosomes with specific cargos that influence pathogenesis. In this study, we searched for proteins relevant to the pathogenesis of PE by analyzing the content of EVT exosomes secreted following nicotine stimulation.

## 2. Results

### 2.1. Effects of Nicotine in a PE Mouse Model

The systolic BP of CD40L control mice gradually increased during pregnancy and remained elevated (120 ± 6 mmHg, *n* = 4) through E17.5, while nicotine-treated mice had significantly lower systolic BP (107 ± 4 mmHg, *n* = 5, *p* < 0.05) at E17.5 (Figure 2A).

Birth weights of pups were lower in the CD40L control group (1.31 ± 0.41 g, *n* = 9) than the nicotine-treated group (1.46 ± 0.35 g, *n* = 7), and placental weight was slightly lower in the control group (0.23 ± 0.03 g, *n* = 9) compared with nicotine-treated mice (0.26 ± 0.03 g, *n* = 5), though these results did not reach statistical significance (Figure 2B). The urinary albumin/creatinine ratio was lower in nicotine-treated mice (0.08 ± 0.04 g, *n* = 5) than control mice (0.35 ± 0.11 g, *n* = 4) (Figure 2C).

### 2.2. Proteomic Analysis and Bioinformatic Characterization

Our proteomics analysis identified and qualified 2477 proteins that we classified and annotated through several bioinformatics approaches. Using GO analysis, the proteins were organized by cellular component (Figure 3A), biological process (Figure 3B), and molecular function (Figure 3C). Using PCA classification, we found clustering with high variance (Figure 4) between the nicotine-treated and control groups. The nicotine group (pink) strongly expressed the characteristics of PC1 (36%), while the control group (orange) expressed the characteristics of PC2 (21%). The two groups were clearly separated, indicating that the protein composition of cultured EVT exosomes is strongly affected by nicotine. Using a volcano plot (Figure 5), we identified 138 proteins (Table 1) that showed statistically significant and large-magnitude differences between the nicotine-treated and control groups (Figure 5). Because the 138 proteins whose expression levels were significantly changed after nicotine stimulation are thought to include several proteins that suppress the pathogenesis of PE, we attempted to extract relevant proteins using network analysis techniques.

### 2.3. Protein Network Analysis

The STRING database were used to analyze protein–protein interactions, and a network consisting of 42 nodes and 60 edges was found (Figure 6). String (pathway) analysis is used to identify proteins that are significantly differentially expressed between treatment groups and that may have biological relevance via relationships with a protein network, rather than functioning in isolation. Using String analysis, we identified 52 proteins distributed across three related groups, and then used topological analysis with a Cytoscape Network Analyzer to identify central hub proteins within the biological network of each group (Figure 7). From this analysis, we identified eight proteins with a high degree, which indicates network hub centrality, and further narrowed this list to five proteins: apolipoprotein A1 (APOA1); ceruloplasmin (CP); kininogen-1 (KNG1); lumican (LUM); and KH domain-containing, RNA-binding, signal transduction-associated protein 1 (KHDRBS1). These proteins have the potential to influence PE pathogenesis based on their annotated biological properties and their differential abundance in nicotine-treated EVT exosomes.

## 3. Discussion

Our study suggests that nicotine might influence pregnancy by altering the expression of cargo proteins in the EVT-derived exosomes; such proteins are known to affect the microenvironment of the decidual layer during implantation and the subsequent remodeling of spiral arteries [7]. Nicotine might inhibit the pathogenesis of PE [17], and we tested nicotine stimulation in a mouse model of PE. Nicotine reduced blood pressure while having a slight effect on proteinuria, although this difference was not significant. The hub proteins extracted from nicotine-stimulated EVT-derived exosomes in this study might influence the pathogenesis of PE.

Nicotine is a critical component of tobacco smoke and a signaling molecule with diverse roles in the central nervous system and vasculature, and nicotine is a major contributor to cardiovascular disease. Nicotine can either stimulate or inhibit ganglia (depending on dose) [18] and binds with nicotinic acetylcholine receptors (nAChRs) [18,19] on several cell types to induce either vasoconstriction (when binding receptors on vascular endothelial cells) or vasodilation (when binding receptors on vascular smooth muscle cells). Furthermore, nicotine binding to nAChRs on endothelial cells might stimulate the secretion of proangiogenic growth factors such as VEGF, thus promoting endothelial cell motility and proliferation and restoring damaged endothelial function [20,21]. In the vascular endothelium, nicotine inhibits the release of the vasoconstrictor ET-1 [22] as well as the vasodilators nitric oxide (NO) and prostacyclin [23,24]. In vascular smooth muscle cells, nicotine promotes vasoconstriction by amplifying the response to norepinephrine and by upregulating the expression of ET-1 receptors (ETA and/or ETB) [25]. Thus, nicotine might promote the pathogenesis of PE via acting on vasoconstriction, but nicotine also exerts neuroprotective effects via nAChRs [12] and can suppress the onset of PE in pregnant women [11]. Furthermore, Mimura et al. reported that nicotine induces the production of placental growth factor (PlGF), which promotes angiogenesis for proper placentation [26]. Though nicotine may carry cardiovascular risks, it also suppresses the pathogenesis of PE; therefore, we focused on nicotine-induced proteins that suppress the vascular damage and inflammatory immune responses that are characteristic of the pathophysiology of PE.

Exosomes can promote cancer growth and metastasis by forming a microenvironment around cancer cells that is protective against genotoxic stress-induced cell death [27]. The cargo within these exosomes can create conditions favorable for tumor growth when acting on local tissues and promotes metastasis when transported by exosomes to distant organs. During pregnancy, EVTs take on a role analogous to that of a cancer cell by secreting exosomes that create a microenvironment protective against autoimmunity and inflammation and supportive of EVT proliferation at the implantation site, placenta, and in distant organs.

In this study, we identified and quantified 2477 proteins from the exosomes of nicotine-treated EVTs. Exosomes in the placenta are mainly released from the trophoblast and are thought to influence endometrial function and create a suitable placental environment [27]. Many of these proteins have predicted cytoplasmic localization, and 1118 proteins are involved in cellular processes (e.g., transcription, DNA replication, and DNA repair) that are common in the trophoblast cell lineage. The majority of the proteins are predicted to have a molecular binding function. Su et al. reported that proteins in the trophoblast-derived exosome are enriched in immune and endocrine functions [28], though the five hub proteins that we identified as differentially abundant in nicotine-treated EVT exosomes have functions related to cell proliferation and invasion. Furthermore, APOA1, CP, and KHDRBS1 may inhibit remodeling defects of spiral arteries.

Within the proteome of nicotine-stimulated EVT exosomes, we identified five hub proteins with potential relevance to PE pathogenesis; we classified these proteins functionally into those predicted to suppress inflammation (APOA1, CP, and KHDRBS1). In PE, the inhibition of EVT proliferation and invasion may suppress remodeling of the spiral artery, resulting in the inhibition of placentation. Therefore, KNG1 and LUM, which promote cell proliferation and invasion, might inhibit the pathogenesis of PE.

KNG1 is the precursor protein for high molecular weight kininogen, low molecular weight kininogen, and bradykinin; these factors are essential for blood coagulation and the construction of the kallikrein-kinin system. Of those three proteins, bradykinin stimulates vascular endothelial cells to produce NO, which relaxes blood vessels and lowers blood pressure. Furthermore, bradykinin may influence placentation and obstruct the pathogenesis of PE by promoting the proliferation and invasion of trophoblasts [29,30]. Thus, the production of bradykinin from KNG1 might suppress the pathogenesis of PE not only by relaxing blood vessels but also by promoting the proliferation and invasion of trophoblast cells. On the other hand, LUM is an extracellular matrix protein associated with signal transduction in cancer cells and can have either pro- or anti-tumorigenic effects in different cancer types [31]. LUM is involved in cellular processes associated with tumorigenesis, including the epithelial-to-mesenchymal transition, cellular proliferation, migration, invasion, and adhesion [32]. Therefore, LUM might also suppress the pathogenesis of PE by promoting the proliferation and invasion of trophoblast cells.

The pathogenesis of PE promotes immune and inflammatory responses, thereby inhibiting the immune tolerance necessary to maintain a normal pregnancy. Therefore, suppression of that inflammatory response might improve placentation and reduce the development of PE by inhibiting EVT apoptosis and indirectly promoting cell proliferation and invasion, thereby promoting remodeling of the spiral arteries. The second class of key extracted hub proteins includes those that suppress immune and inflammatory responses: APOA1, which suppresses the cytotoxic effect of TNFα; CP, which suppresses ·OH, a strong cytotoxic factor in the placenta; and KHDRBS1, which promotes NF-κB-mediated anti-apoptotic activity, as hub proteins. APOA1 is the major protein component of high-density lipoprotein (HDL), which is synthesized in the liver and small intestine. It is associated with cholesterol transport, lipid-cholesterol binding, and lecithin cholesterol acyltransferase (LCAT) activation. LCAT may activate HDL remodeling protein [33]. In PE, the proinflammatory cytokine TNFα is elevated from early pregnancy and may be involved in the pathogenesis of the disease. APOA1 can suppresses this effect and might prevent TNFα from damaging EVTs [34]. CP is the major copper transport protein in the blood, influences iron metabolism, and may scavenge reactive oxygen species, though much is still unknown about the exact function of this protein [35]. However, CP produced by IFN-γ-stimulated monocytes promotes the ferroxidase activity that converts ferrous iron (Fe2+) to ferric iron (Fe3+). This activity inhibits the Fenton reaction between ferric iron and hydrogen peroxide responsible for harmful hydroxyl radicals. In the PE placenta, endothelial cells and trophoblast cells are damaged by increased reactive oxygen species (ROS) [5], and the suppression of ROS by CP may suppress the pathogenesis of PE. KHDRBS1 is a protein implicated in selective splicing, cell cycle regulation, RNA 3′-end formation, and tumorigenesis. KHDRBS1 regulates the nuclear-to-cytoplasmic signaling that activates NF-κB proteins in response to DNA damage. A deficiency in KHDRBS1 reduces this signaling and dampens NF-κB-mediated anti-apoptotic gene transcription, thus promoting cell death [36]. Nicotine can increase KHDRBS1 production by EVTs, suggesting that it inhibits cell death by inhibiting apoptosis. In addition, its predicted function in promoting cell proliferation and invasion suggests that it may alter the proliferation of trophoblast ectoderm cells and promote placentation in PE [34]. These three hub proteins can suppress the inflammatory response. EVT proliferation and invasion are inhibited by cellular damage resulting from an elevated inflammatory response in the decidual layer and the placenta of PE. Disturbed remodeling of spiral arteries in PE might be improved by these three hub proteins stimulating EVTs proliferation and invasion by suppressing the local inflammatory response.

The pathogenesis of PE entails aberrant placentation, and healthy placentation requires the successful remodeling of the spiral artery, which requires appropriate proliferation and invasion of EVTs. Placenta-derived exosomes might be involved in the maintenance of normal pregnancy through maternal-fetal tolerance [37]. Nicotine can stimulate the migration and invasion of the esophageal squamous carcinoma cell line [38], suggesting that nicotine may also promote these processes in EVTs. Impaired remodeling of the spiral artery can result from acute or chronic inflammation early in pregnancy, and this inflammation can be driven by cytokines and ROS. Increased placental ROS is promoted by the release of syncytiotrophoblast microvesicles (STBM) derived from the PE placenta [39,40]. These STBMs can stimulate the release of proinflammatory cytokines from monocytes, resulting in an altered maternal systemic inflammatory response in PE [41,42]. In contrast, nicotine may reduce proinflammatory cytokine release from the placenta in an LPS-induced PE mouse model [43,44]. APOA1, CP, and KHDRBS1 might suppress inflammatory responses in this PE mouse model. Kawashima et al. suggested that nicotine might influence the pathogenesis of PE by increasing PlGF producion by EVTs [45]; serum PlGF levels decline in PE prior to the onset of clinical symptoms [46]. There are likely many other mechanisms by which nicotine affects the pathogenesis of PE.

Though it is generally accepted that smoking may suppress the onset of PE [11,47], and this benefit is observed in Western populations, it is not seen in Asian populations [48]. This suggests that PE risk is also influenced by genetic predispositions and that tobacco smoke and nicotine may have complex and contradictory influences on the pathogenesis of PE. For example, although nicotine can promote the migration and invasion of some cell types, it can also inhibit EVT invasion by downregulating CXCL12 expression via nAChR [49].

In this study, we chose five hub proteins from nicotine-stimulated EVT-derived exosomes that may influence the pathogenesis of PE (Figure 8). APOA1 can obstruct the activity of TNFα, which is increased in the serum of PE patients during early pregnancy [50]. CP can suppress the elevated ROS production of the PE placenta and may influence vascular endothelial damage and trophoblast damage in the placenta [5]. KNG1 can lower blood pressure via vasodilation, and KHDRBS1 may stimulate placentation by inhibiting the secretion of proinflammatory cytokines and promoting cellular proliferation and invasion. LUM may promote the poor placentation associated with PE by inhibiting cellular proliferation.

Our study is limited because we evaluated the effects of nicotine and not smoking itself. Therefore, many of the other effects of smoking were not evaluated.

## 4. Materials and Methods

### 4.1. Animal Experiments with a PE Mouse Model

We examined the effects of nicotine administration on PE-relevant physiological parameters using our previously developed mouse model [51]. Briefly, pregnant Imprinting Control Region (ICR) mice (age 8–12 weeks; CLEA Japan, Tokyo, Japan) were sacrificed and the blastocysts were retrieved from the uterine horns. The blastocysts were infected with adenoviral vectors encoding the human CD40L gene (Ad-CD40L; kindly provided by Dr. Fukushima, Eisai Co., Ltd., Tokyo, Japan) and then transferred into the uterine horns of pseudopregnant ICR mice. (−)-Nicotine hydrogen tartrate salt (Sigma-Aldrich, St. Louis, MO) was administered to mice by osmotic pump (Model 2002, ALZET Osmotic Pumps, Cupertino, CA) at a rate of 3 mg/kg/day (CD40L + saline *n* = 6, CD40L + nicotine *n* = 5). Maternal systolic blood pressure (BP) was measured each morning in triplicate by the tail-cuff method (BP-98E, Softron Co. Ltd., Tokyo, Japan) from embryonic day 8.5 (E8.5) through the day of delivery (E17.5). The mice were euthanized on E17.5 and live pups were weighed. All procedures were approved by the Animal Care and Use Committee of Ehime University (05HE36-16).

### 4.2. Cell Culture

The human EVT cell line HTR-8/SVneo was provided by Dr. Charles H. Graham (Queen’s University, Kingston, ON) and cultured at 37 °C and 5% CO_2_ in phenol red-free Dulbecco’s Modified Eagle Medium (DMEM, Thermo Fisher Scientific, Waltham, MA, USA) supplemented with 10% fetal bovine serum (FBS), penicillin-streptomycin solution (Fujifilm Wako Pure Chemical Corp, Osaka, Japan), 1% L-glutamine (Thermo Fisher Scientific), 1 mM sodium pyruvate (Thermo Fisher Scientific), and 1% non-essential amino acids (Thermo Fisher Scientific).

### 4.3. Extraction of Exosomes

Subconfluent cultures were exchanged into DMEM with exosome-free FBS (Exo-FBS, System Biosciences, Palo Alto, CA, USA) with or without 100 µM nicotine and incubated for an additional 24 h. Culture supernatants were collected and clarified by centrifugation at 3000× *g* for 15 min. The cleared supernatants were amended with ExoQuick-TC (System Biosciences) and incubated overnight at 4 °C to precipitate exosomes which were then collected by centrifugation at 1500× *g* for 30 min. The surface protein composition of the extracted exosomes was analyzed with an Exo-check exosome antibody array (Thermo Fisher Scientific), as shown in Appendix A. After confirming the high purity and proper exosomal surface protein expression (CD81, CD63, and TSG101) of the preparation, the exosomes were resuspended in 50 µL of Exosome Resuspension Buffer for proteomic analysis.

### 4.4. Proteomics Sample Preparation

After cleanup of the sample with cold acetone (1:8 *v*/*v*, 2 h incubation at −20 °C), the protein fraction was sonicated in a lysis buffer (100 mM Tris, 0.5% sodium dodecanoic acid), quantitated using a BCA Assay Kit (Thermo Fisher Scientific), and adjusted to a total protein concentration of 1 µg/µL in the lysis buffer. Following S-S bond cleavage and alkylation of cysteine residues, proteins were digested into peptides (with 400 ng Lys-C, 400 ng trypsin, 37 °C overnight), desalted on a C18 spin column, dried in a centrifugal evaporator, and redissolved by sonication in sample buffer (3% acetonitrile with 0.1% formic acid, *v*/*v*) to a final peptide concentration of 200 ng/µL (as determined by BCA assay).

### 4.5. Nano LC-MS Analysis

Peptide samples were analyzed by nano LC-MS/MS (UltiMate 3000 RSLCnano LC System, Thermo Fisher Scientific) connected to a mass spectrometer (Q Exactive HF-X, Thermo Fisher Scientific). Samples were injected as 400 ng total peptide in 2 μL of sample buffer, and the nano-LC column (2.7 µm, 250 × 0.075 mm, 100 A) was run at a flow rate of 100 nL/min with the outflow monitored by an integrated emitter (CAPCELL CORE MP C18, New Objective Inc, Littleton, MA). An elution protocol of solvent A (0.1% formic acid, *v*/*v*), and solvent B (80% acetonitrile, 0.1% formic acid, *v*/*v*) was implemented with a gradual gradient from 1 to 39% solvent B over 82 min, followed by a steeper gradient of 39–80% solvent B over 14 min. As peptides eluted from the column and electrospray source, MS1 scans were acquired in the Orbitrap over the mass range 495–865 *m*/*z* at 120,000 resolution, followed by MS2 at 30,000 resolution.

### 4.6. Statistical Analysis

Physiological data are represented as means ± standard errors. We tested for statistical significance with the one-way analysis of variance using SPSS software (v27.0.0, IBM SPSS Statistics, Chicago, IL, USA). Differences were considered significant at *p* < 0.05.

### 4.7. DIA (Data Independent Acquisition) Proteomic Analysis

The LC-MS data were processed using Scaffold DIA (v2.1.0, Proteome Software Inc, Portland, OR, USA), and we extracted only those proteins with an FDR < 1%. We used a *t*-test to compare exosome protein content between the nicotine-treated and control groups. We considered *p* < 0.05 to be statistically significant.

### 4.8. Bioinformatic Analysis

The proteins were annotated and classified using the Gene Ontology (GO) resource. Principal component analysis (PCA) was processed by Scaffold DIA. The interaction network of the proteins was analyzed by STRING (v11.0, String Consortium, https://string-db.org/). Cytoscape (v3.8.0, Cytoscape Consortium, https://cytoscape.org/) was used to identify hub proteins in the network by determining Degree Centrality and Between Centrality in the protein network.

## 5. Conclusions

Nicotine markedly alters the abundance of several proteins in EVT-derived exosomes, suggesting a mechanism by which tobacco smoke might influence the pathogenesis of PE. However, the physiological concentration of nicotine that results from tobacco smoking is lower than that used in this study, and cigarette smoke contains many other substances other than nicotine. Therefore, the results of this study may not be immediately useful in clinical practice. However, further studies may reveal the effects of nicotine on the pathogenesis of PE and identify proteins that can treat and prevent PE. In addition, exosome miRNAs can be examined in the context of PE [52,53].

## Figures and Tables

**Figure 1 ijms-24-11126-f001:**
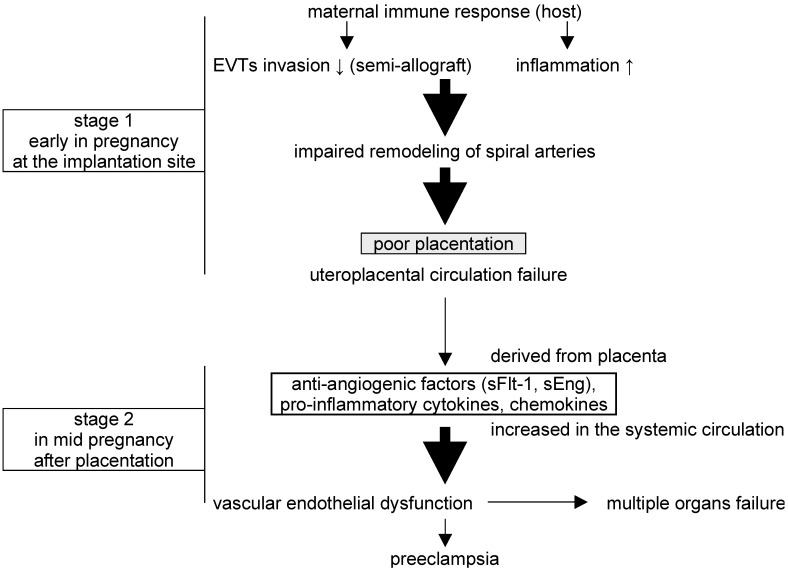
The two-stage theory of the pathogenesis of preeclampsia (PE). In stage one, abnormal placentation results from impaired EVT invasion and reduced immune tolerance for the fertilized egg. In stage two, the abnormal placenta produces anti-angiogenic factors and pro-inflammatory cytokines, resulting in multi-organ vascular dysfunction and the development of PE. Some quotations from ref. [5], modified and added.

**Figure 2 ijms-24-11126-f002:**
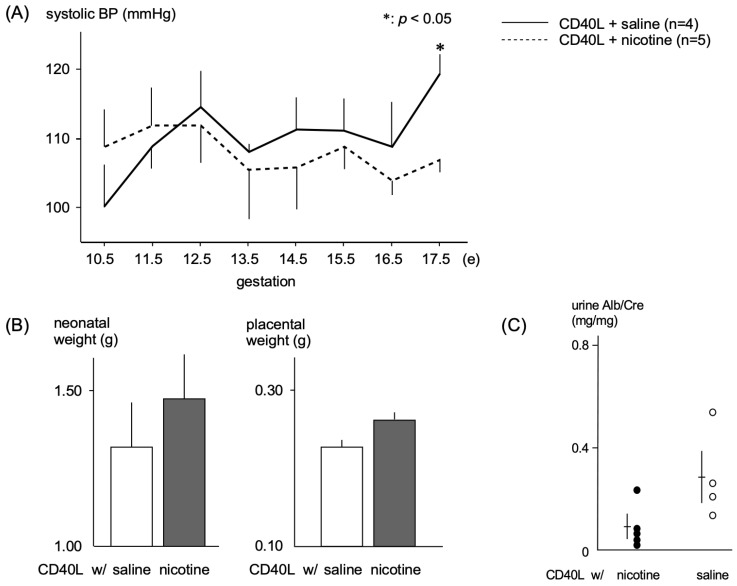
Physiologic effects of nicotine administration in PE model mice (**A**) Maternal systolic blood pressure during gestation. (**B**) Neonatal weight and placental weight of pups born from PE model mice. We observed a mild increase in neonatal weight and placental weight with nicotine; however, the difference was not significant. (**C**) Urinary albumin/creatinine ratio. * *p* < 0.05.

**Figure 3 ijms-24-11126-f003:**
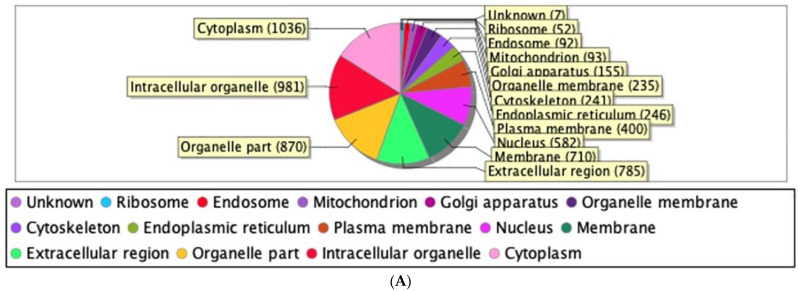
(**A**) Classification of EVT-derived exosome proteins organized by cellular component. (**B**) Classification of EVT-derived exosome proteins organized by biological process. (**C**) Classification of EVT-derived exosome proteins organized by molecular function.

**Figure 4 ijms-24-11126-f004:**
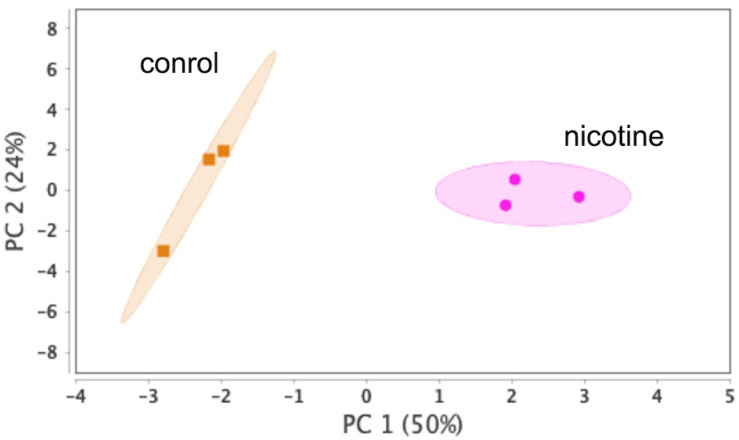
PCA score distribution of EVT-derived exosome proteins.

**Figure 5 ijms-24-11126-f005:**
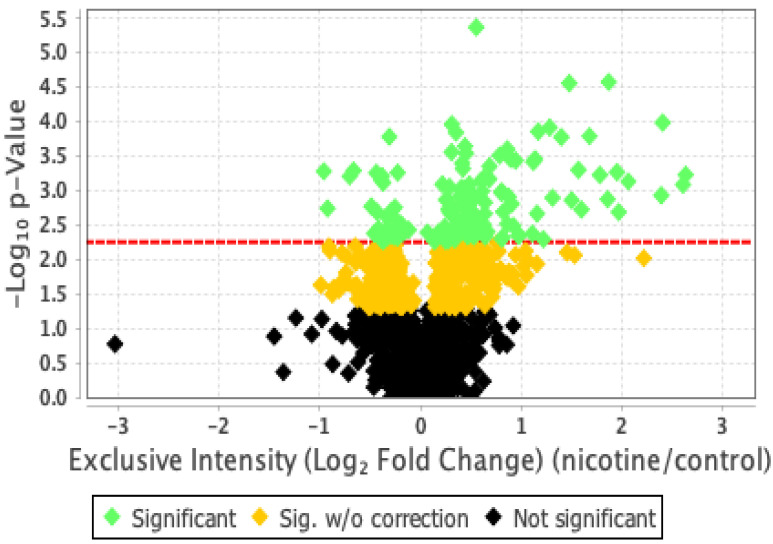
Classification of EVT-derived exosome proteins by *p*-value and differential abundance.

**Figure 6 ijms-24-11126-f006:**
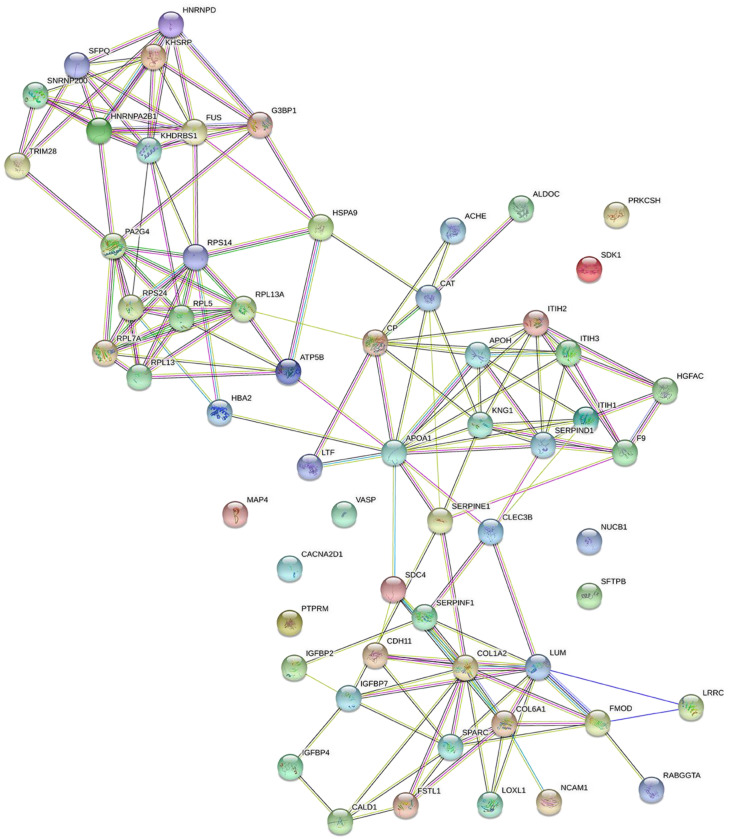
Protein–protein interaction network analysis of EVT-derived exosome proteins.

**Figure 7 ijms-24-11126-f007:**
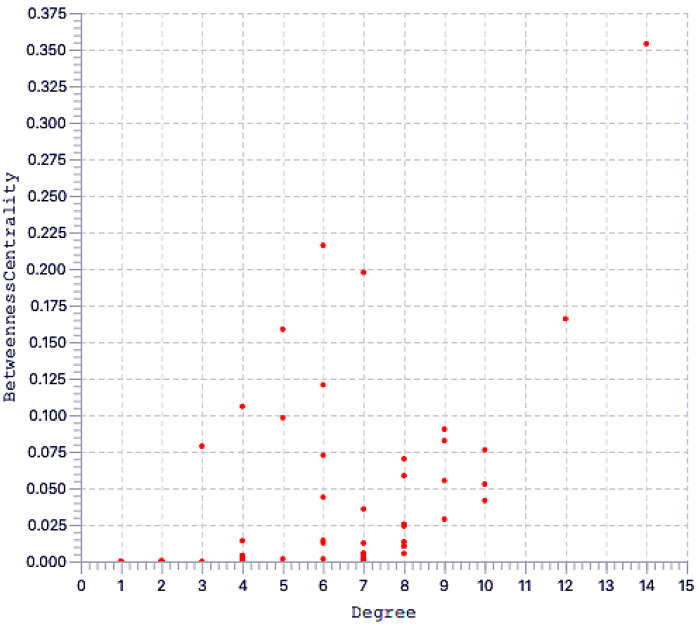
(**Top**) Correlation between degree and betweenness centrality in EVT-derived exosome proteins and (**Bottom**) extracted key proteins.

**Figure 8 ijms-24-11126-f008:**
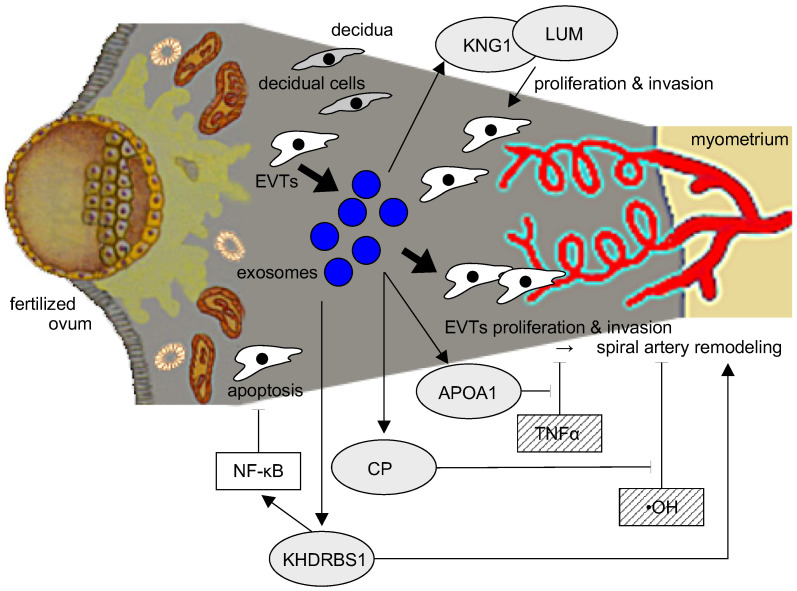
A model of the potential relationships between the five key proteins from our analysis and their influence on PE and EVTs. Kininogen, lumican, and KHDRBS1 are proteins that are more abundant in nicotine-treated EVT exosomes. These proteins promote cell proliferation and invasion, and KHDRBS1 further inhibits cell apoptosis via the activation of NF-κB signaling. APOA1 inhibits TNFα, and ceruloplasmin suppresses -OH and may rescue pathophysiology by improving cell proliferation and invasion of EVT in PE.

**Table 1 ijms-24-11126-t001:** Selected list of top 10 proteins identified with up- and down-regulated log2FC in nicotine-treated EVT-derived exosomes.

Downregulated			
Accession Number	Protein Name	log2FC	*p* Value
Q9UGI8	Testin	2.67	*p* < 0.05
Q02818	Nucleobindin-1	2.63	*p* < 0.001
P13591	Neural cell adhesion molecule 1	2.60	*p* < 0.001
P02788	Lactotransferrin	2.40	*p* < 0.0005
P28827	Receptor-type tyrosine-protein phosphatase mu	2.39	*p* < 0.005
P05114	Non-histone chromosomal protein HMG-14	2.31	*p* < 0.01
P24592	Insulin-like growth factor-binding protein 6	2.23	*p* < 0.05
P55285	Cadherin-6	2.21	*p* < 0.01
Q05682	Caldesmon	2.06	*p* < 0.001
O00533	Neural cell adhesion molecule L1-like protein	2.00	*p* < 0.005
**Upregulated**			
**Accession Number**	**Protein Name**	**log2FC**	***p* Value**
Q6GTS8	N-fatty-acyl-amino acid synthase/hydrolase	−2.08	*p* < 0.0005
Q7Z7L7	Protein zer-1 homolog	−2.00	*p* < 0.05
Q14139	Serine/threonine-protein phosphatase 6 catalytic subunit	−1.78	*p* < 0.0005
Q14997	Proteasome activator complex subunit 4	−1.66	*p* < 0.05
Q03001	Dystonin	−1.51	*p* < 0.05
O60341	Lysine-specific histone demethylase 1A	−1.42	*p* < 0.0001
P68871	Hemoglobin subunit beta	−1.35	*p* < 0.05
P40306	Proteasome subunit beta type-10	−1.34	*p* < 0.05
Q9ULHO	Kinase D-interacting substrate of 220 kDa	−1.26	*p* < 0.001
Q13033	Striatin-3	−1.15	*p* < 0.005

## Data Availability

Not applicable.

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
