# Peer review of "The Influence of Nicotine on Trophoblast-Derived Exosomes in a Mouse Model of Pathogenic Preeclampsia"

_ijms, 2023, doi:10.3390/ijms241311126_

Round 1

Reviewer 1 Report

Thank you very much for allowing me to review the article entitled "The influence of nicotine on trophoblast-derived exosomes in a mouse model of pathogenic preeclampsia" (ijms-2437772). This manuscript is submitted for the "Molecular Pathology, Diagnostics, and Therapeutics" section of the Special Issue "Molecular and Cellular Research in Pregnancy-Related Complications."

Preeclampsia is a serious complication of pregnancy. However, it has been observed that women exposed to tobacco have a lower frequency of preeclampsia. Tobacco is clearly described as a risk factor during pregnancy, but the reason why it protects against the development of preeclampsia is not clarified. This study proposes a possible explanation for this phenomenon.

Comments:

The structure of the abstract allows for the anticipation of fundamental information in the article. In addition, in many cases, when proposing new studies, the abstract is used as a selection criterion. For all these reasons, I believe that the structure of the abstract should be improved. It should not only present the justification of the work but also clearly define the objective, the design, the methodology applied, and the main results and conclusions.

The introduction should be expanded to provide more information and hypotheses regarding the pathophysiological role of nicotine on EVTs (extravillous trophoblasts) to modify exosome secretion and cargo composition in the context of preeclampsia pathogenesis. Additionally, the objective of the study is usually stated at the end of the introduction. Please clarify whether Figure 1 is an original creation by the authors or if it has been taken from another source, in which case the source should be cited.

Results: I don't believe that lines 68-70 are necessary as they are repetitive. Simply indicating the section structure is sufficient. In Figure 2, B, please indicate whether there is a significant difference or not.

Figure 3A and B have overlapping labels in the pie chart. Please try to clarify this figure. The section "Proteomic analysis and bioinformatic characterization" should be interpreted in relation to preeclampsia.

Please provide a better explanation of Figures 6 and 7. Are there significant differences? What utility do they have in understanding how nicotine acts in the prevention of preeclampsia?

Discussion: Much of the information presented in the discussion should be included in the introduction. In the discussion, the main results derived from the experimental study, their biological plausibility, and their strengths and weaknesses to be addressed in future studies should be presented. However, the discussion seems more like a review. Therefore, I suggest reconsidering the discussion.

Materials and methods: Please provide a better explanation of the number of animals and the samples collected over time. It seems that you have almost the same animals at different time points in the study's follow-up. Has the sample size been calculated? Has the normality of the distribution been assessed to determine whether to present the median or the mean? What has been used in the comparison? Do your conclusions truly stem from your results?

Author Response

First of all, I would like to thank the reviewers for their efforts. Each of them provided valuable input. We would like to respond sincerely to their comments.

#1. “The structure of the abstract allows for the anticipation of fundamental information in the article. In addition, in many cases, when proposing new studies, the abstract is used as a selection criterion. For all these reasons, I believe that the structure of the abstract should be improved. It should not only present the justification of the work but also clearly define the objective, the design, the methodology applied, and the main results and conclusions.”

Ans1. Thank you for the comment. We reconstitute the abstract following the comment.

#2. “The introduction should be expanded to provide more information and hypotheses regarding the pathophysiological role of nicotine on EVTs (extravillous trophoblasts) to modify exosome secretion and cargo composition in the context of preeclampsia pathogenesis. Additionally, the objective of the study is usually stated at the end of the introduction. Please clarify whether Figure 1 is an original creation by the authors or if it has been taken from another source, in which case the source should be cited.”

Ans2. Thank you for the comment. We reconstitute the introduction following the comment.

#3. Results: I don't believe that lines 68-70 are necessary as they are repetitive. Simply indicating the section structure is sufficient. In Figure 2, B, please indicate whether there is a significant difference or not.

Ans3. Thank you for the comment. We modified the part of results following the comment. Sorry for the lines 68-70. We forgot to erase the template text.

#4. Figure 3A and B have overlapping labels in the pie chart. Please try to clarify this figure. The section "Proteomic analysis and bioinformatic characterization" should be interpreted in relation to preeclampsia.

Ans4. Thank you for the comment. We made a mistake about the figure. We replaced figure 3B with the correct one.

#5. Please provide a better explanation of Figures 6 and 7. Are there significant differences? What utility do they have in understanding how nicotine acts in the prevention of preeclampsia?

Ans5. Thank you for the comment. We modified the sentences about the fig.6 & 7. Of course, we chose statistically significant data only. We do not believe that nicotine is important for PE prophylaxis just because nicotine suppresses the onset of PE, but rather to understand the pathogenesis of PE by understanding what part nicotine suppresses in the pathogenesis of PE, and to search for proteins that will be effective in the prevention and treatment of PE in the future. The goal is to find proteins that are effective in the prevention and treatment of PE in the future. Therefore, the five proteins discovered in this study will play an important role in the pathogenesis of PE in the future.

#6. Discussion: Much of the information presented in the discussion should be included in the introduction. In the discussion, the main results derived from the experimental study, their biological plausibility, and their strengths and weaknesses to be addressed in future studies should be presented. However, the discussion seems more like a review. Therefore, I suggest reconsidering the discussion.

Ans6. Thank you for the comment. Since the purpose of this study is not to investigate the mechanism of nicotine on PE pathogenesis, but to find important proteins that are altered when nicotine affects the pathogenesis of PE, the discussion is somewhat review-oriented. However, we have modified the discussion of this manuscript as much as possible in accordance with the comment.

#7. Materials and methods: Please provide a better explanation of the number of animals and the samples collected over time. It seems that you have almost the same animals at different time points in the study's follow-up. Has the sample size been calculated? Has the normality of the distribution been assessed to determine whether to present the median or the mean? What has been used in the comparison? Do your conclusions truly stem from your results?

Ans7. Thank you for the comment. We provided the number of the animals in the section of Materials and methods. The normality of the data has been confirmed in a previously published paper (Matsubara K, Matsubara Y. Hypertens res. 2016), and since the normality of the data has been confirmed by graphing the data in this study, there should be no problem in making a comparison using the mean values.

Reviewer 2 Report

The authors propose a work in which they recognize a limitation but explain the reason for their choice, also because the nicotine dosage considered is very high.

The conclusions should be significantly revised as it is not understood what the possible consequences of exposure to nicotine are in practical terms.

We should consider other EV contents, such as microRNAs, that have been shown to have a significant role as extracellular messengers and communicators (10.3390/ph14121257 and 10.3389/fphys.2021.709807)

The conclusions should also be better characterized for possible future studies.

It needs revision.

Author Response

First of all, I would like to thank the reviewers for their efforts. Each of them provided valuable input. We would like to respond sincerely to their comments.

#1. The authors propose a work in which they recognize a limitation but explain the reason for their choice, also because the nicotine dosage considered is very high.

Ans1. Thank you for your comments. The comment is correct in what the reviewer2 said. On top of that, we constructed the methodology of this study to recover clear data by high dose of nicotine.

#2. The conclusions should be significantly revised as it is not understood what the possible consequences of exposure to nicotine are in practical terms.

Ans2. Thank you for the comment. We revised the conclusion.

#3. We should consider other EV contents, such as microRNAs, that have been shown to have a significant role as extracellular messengers and communicators (10.3390/ph14121257 and 10.3389/fphys.2021.709807)

Ans3. Thanks for your comment. We have included it in our conclusions as an item for future consideration and cited the papers you recommended.

#4. The conclusions should also be better characterized for possible future studies.

Ans4. Thanks for your comment. We revised the conclusion.

Round 2

Reviewer 1 Report

Thank you very much for giving me the opportunity to review the revised version of the article titled "The impact of nicotine on trophoblast-derived exosomes in a mouse model of pathogenic preeclampsia" (ijms-2437772), as well as the authors' response to the suggested changes.

This study involves experimental research using animals to assess proteins that are relevant to the development of preeclampsia (PE) by analyzing the content of exosomes secreted by extravillous trophoblasts (EVTs) after stimulation with nicotine.

Exosomes are known to contain a diverse range of molecules, including proteins, RNA (such as messenger RNA, microRNA, and long non-coding RNA), and DNA. It is plausible that, following nicotine stimulation, these exosomes may contain several proteins that can inhibit the progression of PE.

The findings indicate that nicotine could potentially impact pregnancy by modifying the expression of cargo proteins in EVT-derived exosomes. These proteins are known to influence the microenvironment of the decidual layer during implantation and the subsequent remodeling of spiral arteries.

In my opinion, the article has become much clearer after the revision, providing highly relevant information for future advancements in the prevention of pre-eclampsia.

Author Response

Thank you for your reply, revising our manuscript according to the reviewer's comments gave us new insights and made the manuscript better.

Thank you for the peer review.

Reviewer 2 Report

I think that the manuscript has improved.

Just a slight revision is needed.

Author Response

Thank you for your reply, revising our manuscript according to the reviewer's comments gave us new insights and made the manuscript better.
Thank you for the peer review. We will make a further NATIVE CHECK and resubmit.